# A New Perspective on Fluid Simulation: An Image-to-Image Translation Task via Neural Networks

## Abstract

Standard numerical methods for creating simulation models in the field of fluid dynamics are designed to be close to perfection, which results in high computational effort and high computation times in many cases. Unfortunately, there is no mathematical way to decrease this correctness in cases where only approximate predictions are needed. For such cases, we developed an approach based on Neural Networks that is much less time-consuming but nearly as accurate as the numerical model for a human observer. We show that we can keep our results stable and nearly indistinguishable from their numerical counterparts over tenth to hundreds of time steps.

## 1 Introduction

### 1.1 Target Issue

Simulating fluids, streams, and flows is a task in many fields of science. In most cases, this is done by numerical approaches like the method of finite elements (FEM) (Quarteroni & Valli, 2008) or Lattice Boltzmann methods (LBM) (Mohamad, 2011). These methods are providing great advantages like configurable accuracy and fine-grained adaptability regarding the specific given task. However, to get close-to-reality simulations, some issues have to be targeted as well.

One of the most important issues emerges from the approximation property of these methods. This property limits the configuration of accuracy in both directions: if the accuracy becomes too low, the approximation will become to coarse — in other words, it will no longer reflect reality respectively fulfill the given task. In contrast, if it becomes too high, effects like the curse of dimension will set in, causing exploding problem sizes. In the result, these methods will have an only minimal adjustable computation time range, as the accuracy strongly connects to the problem size and therefore also to the computation time, and may exceed every given maximum time span.

To avoid this issue, a different approach might be more satisfying. As our focus is on problems with no need of high accuracy, we decided to follow a more approximate path by involving the human observer. This observer typically isn't interested in the numerical representation of the simulation, but in the easy-to-interpret image-based one. That leads directly to the idea to focus on an image-to-image translation for each time step of the simulation. The results of such an approach will be much more inaccurate as typical numerical results — the very limited resolution of the color spaces alone results in high errors compared to FEM or LBM results — but for a wide class of problems the results will be good enough under the condition that the computations are very fast. Therefore, the main task for this approach is to translate an image representation of time step $n$ into the image representation of step $n + 1$ as fast and as accurate as possible. For translation, we decided to use a neural network based on good results for image-to-image translation (Isola et al., 2017).

In total, this approach — starting with an image representation of the starting values and translate it recursively into the next time steps of the simulation — leads to the following question:

- **Is it possible...** regarding computation time and accuracy. Can we get results fast enough to accept the additional approximation? Is the accuracy high enough to get useful results? Will this accuracy remain over many simulation time steps?

- **... to get accurate simulations ...** regarding the real world and the human observer. How much noise in the picture is too much? When and why does our approach begin to fail?

- **... very fast ...** regarding the computation time and deployed hardware. After training, is it possible to run a complex simulation on my laptop at home? Will it be fast enough to meet real-time conditions?

- **... with this approach ...** regarding the recursive usage of neural networks as well as an unedited image input. Can we use unedited data, or do we need preprocessing steps like FFT? Is the recursive approach expedient? Are image processing steps like morphological filtering needed between each step? Is the cGAN approach with a UNET architecture reasonable? Do we need additional LSTM units?

- **... while using as little as possible parameters?** regarding the input-output-ratio. How much data is needed to produce the same output as LBM or FEM? How generalizable is our approach? Do we need to train every explicit model, geometry, and structure or is it possible to transfer results?

Some emerging questions aren't trivial at all, and this paper can't answer all of them. With our work, we want to show the benefits of the mentioned approach and highlight opportunities and limits.

## 1.2 STATE OF THE ART

As our approach covers multiple topics, this state-of-the-art section will compare our work to numerical approaches, image translation methods, architectures of neural networks, comparable approaches for fluid dynamics, and previous work our approach is based on. For a better overview, we will headline the corresponding sections and summarize the latest developments.

**Numerical Approaches:** Standard numerical methods for solving PDEs, like FEM (Quarteroni & Valli, 2008) or LBM (Mohamad, 2011), are widely known. Ideas like approximate preconditioning (Anzt et al., 2018) or multi-precision solvers of systems of equations (Gratton et al., 2019; Aliaga et al., 2020) are emerging over the last years. These approaches can have a great impact on a specific part of the numerical PDE solver, but this doesn't necessarily lead to faster run-times for the whole PDE solver. There are also some more global approaches to speed up a complete PDE solving method thanks to mathematical optimization, like Gracie et al. (2006); Du & Wang (2015); Etzmuss et al. (2003), but these approaches are highly adapted to a specific problem. Based on these findings, a pure mathematical approach doesn't seem to be the right way.

**Image Translation:** Our basic idea is based on image translation with a neural network. In the last years many approaches in this direction appear, manly (but not only) to manipulate images or movies in real-time (Liu et al., 2017; Radford et al., 2015; Zhao et al., 2020). Most of these approaches are not really matching our task, which results in unsuitable network architectures or unrealizable constraints. But there is one matching approach with Isola et al. (2017). Our cGAN approach is inspired by the ideas and the excellent results given there and in additional work on cGAN structures like Karras et al. (2017); Zhang et al. (2017).

**Neural Network Architectures:** In addition to the cGAN approach, we need to find the right architecture for our neural networks. Based on Isola et al. (2017) we used a PatchGAN architecture (Li & Wand, 2016) for the discriminator part. For the generator part, we used the proposed U-Net architecture (Isola et al., 2017). Regarding the iterative structure of our translation task, we added the idea of long short-term memory modules (LSTM) (Hochreiter & Schmidhuber, 1997) to improve the U-NET structure.

**Neural Fluid Dynamics and PDE Solvers:** Combining numerical methods with machine learning algorithms is not entirely new, even in the field of fluid dynamics. Two very up-to-date examples are Li et al. (2020) and Pfaff et al. (2021). While the first one focuses on the numerical operators and numerical errors, the second one tries to work with the numerical discretization. Both – as other approaches before – are following approaches starting within the numerical solving pipeline. Our approach to work solely on the image representation clearly separates us from previous approaches and is a unique characteristic of this paper. Unfortunately, our work is not yet at the point to be fully comparable with these approaches, but we are aiming to it in the near future.

**Additional Influences:** Finally, there is one more idea we have to mention as part of the base of our work. In Lehmann et al. (2020) the authors showed, why and how a binary map is a good option to define a sharp separation of areas within an image regarding the image translation task. We adopted this option for our approach, as we need a sharp separation between the streaming area and the environment area in the image representation.

### 1.3 CONTENT ORGANIZATION

In detail, we will provide data for the following main findings in this paper:

- Using a pix2pix-approach with a U-NET structure and a cGAN training is a useful way to get approximated simulation results
- Results from recursive application of neuronal networks are useful over tenth to hundreds of iterations, including an only moderate increasing additional approximation error
- The speed-up can be upto 9 (on GPU-based hardware) compared to a FEM-based simulation
- Different color spaces, boundary mappings, and input data sets are possible and may lead to data-driven approaches

We structured this paper as the following: We start with some basic knowledge about FEM and image comparison, followed by explanations of our neuronal network architecture, and our data generation in section 2. Section 3 will cover our results for our test setting (see A.3) and will give a look over the edge of this specific configured environment. We close this work with a conclusion and summary in section 4 as well as a view on our future directions. The statement of r/eproducibility can found at the very end.

## 2 THEORETICAL BACKGROUND

### 2.1 NUMERICAL BASIS AND MESSURMENT

Looking into our approach, everything starts with a given PDE (partial differential equation), the mathematical model behind descriptions of physical phenomenons. In our case, this is the Navier-Stokes equations for incompressible flows (Oymak & Selcuk, 1996). The starting conditions, boundary conditions, and physical parameters are chosen in a way to get a Kármán vortex street within a canal with an obstacle (fig. 1) (Schäfer et al., 1996). For discretization, the method of lines (Oymak

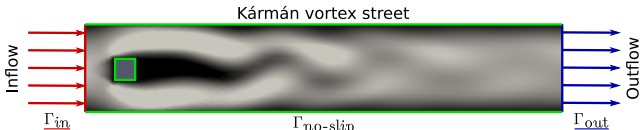

Figure 1: Sketch of the experiment with die inflow boundary condition $\Gamma_{in}$ (inflow speed) , the outflow boundary condition $\Gamma_{out}$ (outflow speed), and with and a no-slip boundary condition $\Gamma_{no-slip}$.

& Selcuk, 1996) is used, which leads to discrete time steps each equipped with the same discrete space grid. The values derived from the chosen solvers[1] are finally mapped to a chosen color space (in our case it is the $[0, 255]$ grayscale).

This very rough spotlight on the basic method shows where the main issue can be located: There are a lot of approximation errors in the reformulation, discretization, numerical solvers, and in the final mapping to the color space (Quarteroni & Valli, 2008). Additionally, the single errors may (or may not) accumulate, which is the reason it can be very difficult to change a single solver or transformation method out of the solving pipeline. Especially in the case of neural networks, where

---

[1]$\Theta$-step methods for the time direction (Berzins & Furzeland, 1992) and a variant of Newton's method (Nemec & Zingg, 2002) for the non-linear equations in space.

no useful formal error limit guaranties can be given, a global replacement approach seems to be the more promising than changing a single pipeline step.

As this means to replace the whole numerical method with a neural network and only keep the starting values to create the first input data, we have to solve the issue to define a meaningful quality measurement for our results. For run time a simple time difference, respectively a speed-up measurement, is good enough. For accuracy, comparing the numerically created image of a time step with one created by a neural network pixel by pixel may lead to high errors, even if the images are indistinguishable by a human observer. However, the same measurement may only result in small errors even if one can find obvious false streaming data (with small changes in the colors). As we found in previous testing, this problem will occur in our case also for average pixel errors and correlation-based measurements. Therefore, the most promising quality measurement for us is based on the mean square error (MSE) and originally developed to evaluate the quality of lossy compression algorithms. It is called peak signal-to-noise ratio (PSNR) (Korhonen & You, 2012):

$$\varepsilon_{\text{PSNR}} = 10 \cdot \log_{10} \left( \frac{255^2}{\varepsilon_{\text{MSE}}} \right) \quad [dB]. \tag{1}$$

In theory, a higher PSNR value means less detectable differences in the images, and values above $30 \, dB$ should result in only undetectable differences for a human observer (Mehra, 2016). In practice, we noticed that in many cases, values below this significant value might be acceptable as well. The reason is that the first observable errors are irrelevant ones regarding our setting, like discolored vertical or horizontal pixel lines that obviously cannot be interpreted as streaming data. Therefore, we marked the theoretical limit in our charts and stopped our iteration some steps later.

## 2.2 BASIC APPROACH AND ARCHITECTURE OF NEURONAL NETWORKS

As mentioned earlier, we see the task of fluid simulation within the calculation of the next time step combined with the representation of the result in a human-readable fashion in a slightly different way. What we see is a representation (image) of a current state of the fluid flow, and the goal is a representation (image) of the next state of the fluid flow. Therefore, in our eyes, this is nothing else than an image-to-image translation task.

Well-known approach for this is pix2pix (Isola et al., 2017). This is a general-purpose solution for image-to-image translation problems based on conditional Generative Adversarial Networks (cGANs). In contrast to GAN, where these networks learn a loss that tries to classify if the output image is real or not and simultaneously train a generative model to minimize this loss, cGANs learn a conditional generative model (Goodfellow et al., 2014).

As explained in Isola et al. (2017), the traditional GAN method uses a random vector $z$ as an input to the generator network $G$ to generate output $y$, $G : z \rightarrow y$. Oppositely conditional GANs additionally feed an input image $x$ to the generator, $G : x, z \rightarrow y$. Isola et al. (2017) and Wang & Gupta (2016) suggest that in certain cases the usage of $z$ can be usefully, but we decided not to include the random vector for our generator, as we want a deterministic network. Fig. 2 illustrates this principle.

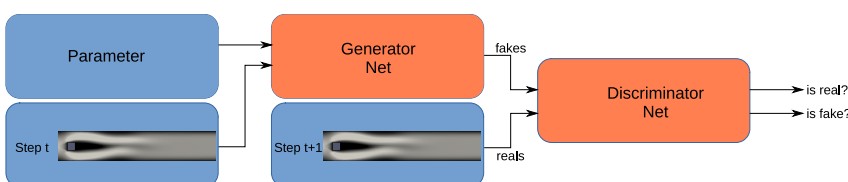

Figure 2: Illustration of the cGAN principle. The generator gets as input the image of the flow at the time step $t$ and some additional parameters like the binary map and the inflow speed.

The discriminator network is modeled with the function $D : x, y \rightarrow v$ that evaluates the likelihood of $y$ being a real image. To note is that the discriminator network has access to the real image $x$ and tries to guess, if $y$ is the real or generated output.

$$\mathcal{L}_{cGAN}(G, D) = \frac{\mathbb{E}[\log D(x, y)] + \mathbb{E}[\log D(x, G(x))]}{2} = \frac{\log D(x, y) + \log D(x, G(x))}{2} \tag{2}$$

where $x$ ($y$) is the input image (target image). The objective function is divided by two to slow down the training of the discriminator relative to the generator, as suggested in Isola et al. (2017). The objective function for the generator network is composed of two parts – the value of the discriminator as well as a $L1$ distance loss between the target and the predicted image. According to Isola et al. (2017) the $L1$ loss promotes less blurring and captures the low frequencies details of the images. The final objective function for the generator is thus:

$$G^* = \arg\min A_G \max_D \mathcal{L}_{cGAN} + \lambda\mathcal{L}_{L1}(G), \qquad \mathcal{L}_{L1}(G) = \mathbb{E}[\|y - G(x)\|_1] \qquad (3)$$

For all models, we used $\lambda = 100$ as in Isola et al. (2017).

### 2.2.1 GENERATOR NETWORK

For the generator network, we oriented us as well on Isola et al. (2017). The basic structure is a U-Net (Ronneberger et al., 2015). It is a standard encoder-decoder model (Hinton & Salakhutdinov, 2006) with skip-connections between parts of the encoder and decoder. The basic approach uses blocks of layers from convolution-normalization-ReLU (Ioffe & Szegedy, 2015). The encoder-decoder first down-samples the input till the bottleneck layer is reached, followed by an up-sampling to the original size of the input image. In appendix in fig 12 one can see the U-Net with eight layers. The dropout layers work with a dropout rate of 50%. We also experimented with LSTM (Hochreiter & Schmidhuber, 1997) blocks between encoder and decoder to see if features extracted from more than the last step are useful.

As input, the net receives a tensor consisting of the concatenated images of the two velocity directions and the binary map of the simulation, which contains the sharp separation between the streaming area and the environment area of the image. In addition, the inflow velocity is passed. After the prediction image is decoded, the binary map is used again to perform a pixel-by-pixel multiplication leading to better preservation of the streaming area (Lehmann et al., 2020).

### 2.2.2 DISCRIMINATOR NETWORK

For the discriminator, we follow the method of Isola et al. (2017) and we used their PatchGAN discriminator network. To note is that the whole image is given as an input. In our case, we decided to go with a patch size of $286 \times 286$ pixels, in contrast to the suggested patch size of $90 \times 90$ pixels in Isola et al. (2017). In appendix in fig. 11 one can see the structure of the discriminator.

## 3 EXPERIMENTS

### 3.1 EXPERIMENT SETUP

To test our approach, we generated the training data by performing numerous simulations of incompressible fluid flow around a rectangular object (see fig. 1). In our case, we have in total three adjustable parameters – inflow speed $s_{in}$, fluid density $\rho$ and fluid kinematic viscosity $v$. The values of the parameter were chosen to investigate the region between laminar and turbulent flow – the so-called Kármán vortex street.

The simulations were performed numerically by *Elmer FEM* (elm, 2021), a numerical solver library. To visualize the results, we used *Paraview* (Ahrens et al., 2005) and exported grayscale images for the velocity in x- and y-direction, as well as grayscale images for the pressure field. In appendix in fig. 13 one can see images of the time steps $t = 0, 25, 100, 250$ with the inflow speed of $s_{in} = 1.5625\frac{m}{s}$. Further details in appendix A.2.

For training, we used 33 random picked simulations. Three of them are used for validation. Eight additional simulations are used for testing the architecture after training. With this, we have a classical 80-20 split for testing and training.
We used the Stochastic Gradient Decent (Kiefer & Wolfowitz, 1952) with the Adam optimizer (Kingma & Ba, 2017) with a learning rate of $0.0002$ and standard momentum parameters $\beta_1 = 0.9$ and $\beta_2 = 0.999$. All models were trained over 45 epochs with a batch size of 3. As programming environment we used *PyTorch* (Paszke et al., 2019) in version 1.8.1. For the testing and training environment, we refer to appendix A.3.To evaluate our experiments, there are two possibilities:

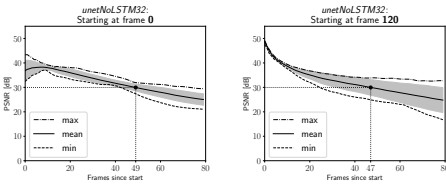

Figure 3: Performance chart of *unetNoL-STM32* with respect to PSNR. The dashed lines are highlighting the value range while the gray area shows the standard deviation around the mean value (solid line). The iteration step where the mean value falls below the value of 30 $dB$ is marked with the dotted line.

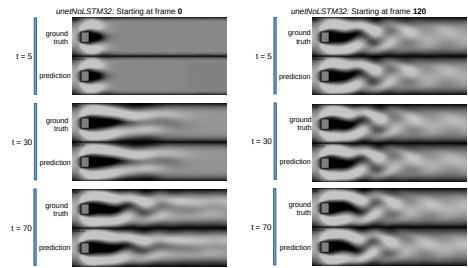

Figure 4: Predictions of *unetNoLSTM32* with starting frame zero and 120 at $t = 5, 30$ and 70 in comparison to the ground truth.

Single-image performance and recursive-application performance. In the first case, only the next time step is calculated and evaluated with the real image. In the case of recursive-application, the calculated result of our architecture is again used directly as input of the next calculation cycle. This can be used to estimate how many future time steps can be predicted before the approach falls below the required PSNR (see chap. 2.1).

## 3.2 RESULTS

We examined a variety of different networks. The structure is briefly explained in each case, following the principle already known from chapter 2.2. The results are then presented in each case regarding the single-image-performance and the recursive-application-performance. In the appendix A.4 you can find the images and charts in a bigger version.

### 3.2.1 GENERATOR WITH EIGHT LAYERS WITHOUT LSTM

This net is the starting point of all following nets. The structure follows fig. 12 with 32 feature maps.The PSNR value is in the single-image-evaluation on average $48.59$ $dB$. More interesting is the recursive application. Since the dataset contains frames where the vortex street is fully developed, but also frames where the vortex street is developing, it is interesting whether our approach can represent both streaming states. Therefore, we decided to evaluate the recursive approach with two different starting frames: 0 and 120. The PSNR score rises in both cases above the desired value of 30 $dB$ in the beginning (see. fig. 3). The second finding is that the performance of our net is better on developing vortex streets. With a start frame of 0, the PSNR value holds from over 30 $dB$ to about 49 future time steps. In contrast, with a start frame of 120 this can be held only for 47 time steps. In fig. 4 one can see the result of our net compared with the ground truth for starting frame zero and 120. A possible reason for better performance for developing vortex streets is that on the right-hand side of the channel, almost no pixel transformation has to be done per recursion step.
From this starting point, we tried to get a better accuracy[2] by using the extracted features of more than one iteration step thanks to an LSTM.

### 3.2.2 GENERATOR WITH EIGHT LAYERS WITH LSTM

In this network, a LSTM block is placed between encoder and decoder. We expect this to provide a better representation of temporal behavior. This results in a single-image PSNR of $48.5$ $dB$ and slightly worse than without the LSTM. In contrast, starting the recursive-application at frame 0 the PSNR starts at a higher value than before (see fig. 5). In the beginning, the standard deviation is smaller too. But after dropping under the line of 30 $dB$ the standard deviation starting to get bigger than the standard deviation without a LSTM. With LSTM, the value can be kept above the required value of 30 $dB$ for 60 time steps and is thus about 11 time steps ($\approx 22\%$) better than before. Starting at frame 120 we see an entirely different picture. The peak value for the first frame is lower

---

[2]In the meaning of generating more iteration steps above the limit of 30 $dB$.

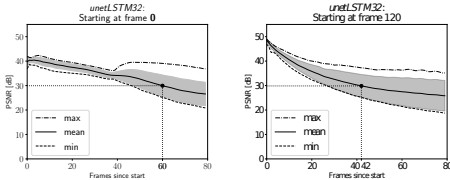

Figure 5: Performance chart of *unetLSTM32* with respect to PSNR. The dashed lines are highlighting the value range while the gray area shows the standard deviation around the mean value (solid line). The iteration step where the mean value falls below the value of 30 $dB$ is marked with the dotted line.

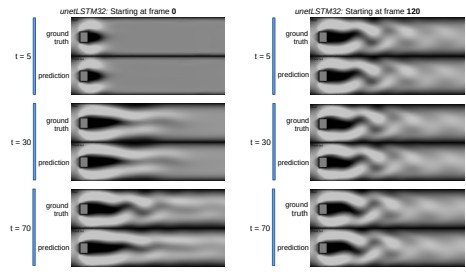

Figure 6: Predictions of *unetLSTM32* with starting frame zero and 120 at $t = 5, 30$ and $70$ in comparison to the ground truth.

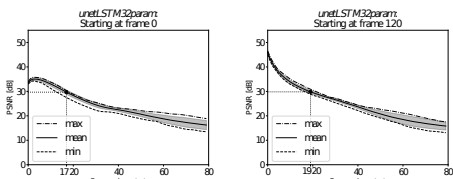

Figure 7: Performance chart of *unetL-STM32param* with respect to PSNR. The dashed lines are highlighting the value range while the gray area shows the standard deviation around the mean value (solid line). The iteration step where the mean value falls below the value of 30 $dB$ is marked with the dotted line.

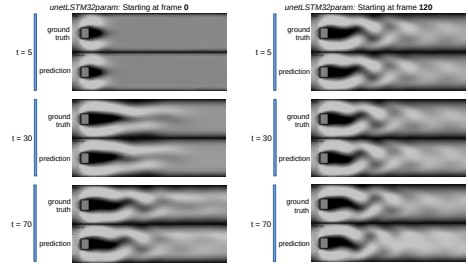

Figure 8: Predictions of *unetLSTM32param* with starting frame zero and 120 at $t = 5, 30$ and $70$ in comparison to the ground truth.

and the standard deviation is significantly higher than without the LSTM-block. Furthermore, the PSNR-score drops early under the 30 $dB$. As before, in fig. 6 one can see the result of this net compared to the ground truth. To keep the good performance for the starting sequence of the stream and optimize the performance for the later stages, we tried to improve the LSTM module next.

### 3.2.3 Generator with eight layers with LSTM with additional input to LSTM

Since the previous LSTM has very restricted access to information, we decided to give it the inflow speed of the fluid as an additional parameter. The single-image performance results in a PSNR score of 45.8 $dB$. The recursive-application starting at frame 0 shows a much worse result than the nets before (see fig. 7). Only about 17 future time steps, the PSNR score can be hold over 30 $dB$. But the standard deviation is significantly lower than before.
Starting at frame 120 the picture is over all the same. The score can be hold over 30 $dB$ about 19 time steps. And the standard deviation is less, too.
In summary, these results are not satisfactory and sufficient. As before, in fig. 8 one can see the result of this net compared to the ground truth. To note are the developing artifacts in front of the object to be flowed around. As simply using an LSTM for the whole iteration process my not be the right way of improving our accuracy, the next step was to build up the network structure in total.

### 3.2.4 Generator with eight layers with LSTM with more feature maps

A final open question is what happens when one invest more computing power. This was realized in the form of more feature maps. The basic structure of the architecture remained the same. However, the number of feature maps per layer was doubled. Because of the good performance in the starting phase, the LSTM was retained between the encoder and decoder in its original, not parameterized form. The charts in fig. 9 shows the result of this decision. In single image evaluation, the average

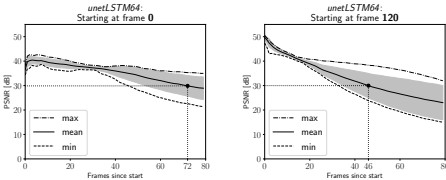

Figure 9: Performance chart of *unetLSTM64* with respect to PSNR. The dashed lines are highlighting the value range while the gray area shows the standard deviation around the mean value (solid line). The iteration step where the mean value falls below the value of 30 $dB$ is marked with the dotted line.

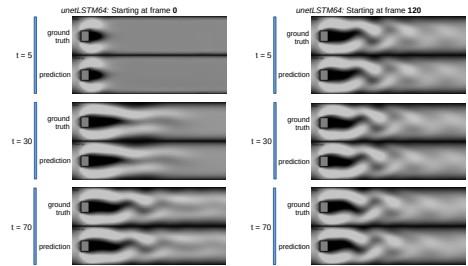

Figure 10: Predictions of *unetLSTM64* with starting frame zero and 120 at $t = 5, 30$ and $70$ in comparison to the ground truth.

PSNR-score is $49.8\ dB$, which is the best score we achieved in our experiments. Similar results are shown in the recursive application. At starting frame 0 the PSNR-score can be held over 30 $db$ for 72 time steps and at starting frame 120 for 46 steps. As before, in fig. 10 one can see the result of this net compared to the ground truth. No artifacts are developing over the time.

### 3.3 EVALUATION

From our results regarding the accuracy, we can conclude, that it is possible to iteratively generate a sequence of streaming data images with an accuracy comparable to the image representations of the numerical solver from the perspective of a human observer. We get the best performances with our given cGAN approach and a U-NET structure with 64 feature maps in the first layer. In the developing phase of the Kármán vortex street, an LSTM is clearly helpful. In the later stages a parameterized LSTM might help to lower the standard deviations, but in total, no LSTM seams to be the better option. Ideally, one would start the simulation using neural networks with an LSTM between encoder and decoder. After a certain number of time steps, the positive influence of the LSTM decreases so that it could be helpful to then switch off the LSTM and benefit from the better performance without LSTM in already developed vortex streets.

Before we take a look into the run times, we want to give a closer look over the edge of the previously given experiments to show some limits, respectively side effects, of this approach.

### 3.4 A LOOK OVER THE EDGE

While developing a variety of different structures, a variety of problems has also arisen. The biggest problem is the creation of artifacts and misleading structures which are not directly visible in the PSNR. For instance, black pixels indicates a velocity of zero. Growing areas of black pixels means growing areas without a flow at all. The problem is that if these artifacts are small, the PSNR stays nearly untouched. With help of the LSTM between encoder and decoder, the arising of black artifact has been greatly reduced.
We also tested our approach with values mapped to an RGB color space. In this case, we don't need the binary mask anymore because black pixels can be set as boundary or objective. But there are other problems, like more artifacts, which negate this advantage. Moreover, the effort of the net structure is multiplied by around 3 because of the additional color channels.
We have also found that our database is relatively small. The Kármán vortex street is very prominent and intentionally always occurs in our data set, which is why an overfitting effect is probably certainly present. To counteract this, we experimented with objects in the center of the channel in both the $x$ direction and the $y$ direction. Thus, it is possible to double the database again by simple mirroring. However, this did not result in improved properties of the mesh. On the contrary, it led to worse results, since the information about the flow direction is lost.Thus, this led to unpredictable events by inverting the flow during a simulation.
As the numerical simulation also returns a pressure field for each step, we tried to use (and generate) it with our approach. We found that in some cases the usage of the pressure field as an additional input improves the accuracy in all stages, in other cases the accuracy decreases. At this moment, we

Table 1: $T_{CPU}$ ($T_{GPU}$) is the time spend on calculating 335 time steps on CPU (GPU). $S_{CPU}$ ($S_{GPU}$) is the speedup on CPU (GPU) with our method compared to the numerical method on CPU.

| Method | $T_{CPU}$ in s | $T_{GPU}$ in s | $S_{CPU}$ | $S_{GPU}$ |
|---|---|---|---|---|
| *unetNoLSTM32* | 123.61 | 14.98 | 1.13 | 9.30 |
| *unetLSTM32* | 129.81 | 15.32 | 1.07 | 9.09 |
| *unetLSTM32param* | 131.51 | 15.39 | 1.06 | 9.05 |
| *unetLSTM64* | 216.70 | 15.51 | 0.64 | 8.98 |
| *Elmer FEM + Paraview* | $42.32 + 96,92$ | - | 1 | - |

cannot clearly separate these cases from each other, and no data can be given here. However, the accuracy of the generated pressure field doesn't seem to differ from the velocity field ones.

Likewise, we experimented with another data set. In this one, we replaced the rectangular object with a round one and further ensured that a vortex street was created. Similar results were obtained with this new data set than with the other data set. This suggests that the architecture is generalizable, but it remains to be verified, however.

## 3.5 CALCULATION TIME COMPARED TO THE NUMERICAL METHOD

In tab. 1 one can see the comparison of the run time of our methods compared to the numerical calculation with *Elmer FEM*. All tests are done on the in section A.3 mentioned system. The CPU versions were executed on all available processor cores. The *Elmer FEM* solver is optimized in this respect, which is also reflected in the runtimes. In this respect, our approach has not yet been optimized for runtime. Thus, the predicted images are written to disk directly after execution and are not buffered or exported in parallel to the next execution. The situation is similar with the multiplication of the mask after prediction. This is also still done in serial execution. Our method benefits of the use of a GPU. As mentioned earlier, we tackle the problem as an image-to-image approach. Thus, as a result, we have images of our flow as a PNG a human observer can immediately interact with. In contrast, the *Elmer FEM* solver produces a non-intuitive data format that must be processed by Paraview (or similar software). In the combined execution with *Elmer FEM* followed by *Paraview*, we achieve a speedup of about 9 with our method using the GPU, which is excellent considering that there was no focus on optimizing the execution time.

## 4 CONCLUSION

### 4.1 CONCLUSION

With this paper, we show an entirely new approach to numerical flow simulation. We consider the aforementioned problem as an image-to-image translation task. Thus, we developed an approach to solve the problem using neural networks, since they have shown strong performance in this particular area in the past. Due to the conversion to images, a direct comparison with the numerical solution is not possible, so we decided to focus on the human observer. Therefore, we set the PSNR as the evaluation metric and showed that there is a lot of potential in this idea. We can answer the question "Is it possible to interpret the numerical flow simulation as an image-to-image translation?" with yes. We also showed that even not fully optimized neural networks can predict up to 72 future time steps without being a significant limitation for the human observer. Without any run time optimization, the new approach also shows a significant speed-up of 9 compared to the numerical method when using the appropriate hardware resources.

### 4.2 FUTURE WORK

Based on the shown results, we see a lot of work for the future. A lot of the questions shown in the introduction are still unanswered. Our network architecture has a lot of optimization potential regarding the prediction accuracy. The iteration process isn't optimal regarding the runtime. Our test setting is very limited at the moment in every dimension: geometry, structure, basic model, parameter settings, and so forth. Fortunately, our results are promising for these questions and issues, and we see great potential in our presented approach.

## 5 STATEMENT OF REPRODUCIBILITY

For reproducibility, we tested our approach on different systems. As mentions in appendix A.3 we run it on an HPC and our local server. Training, Testing and Evaluation was done for reproducibility proposes on both system. For time saving and comparability issues, we choose the mentioned setup. Furthermore, we tested it on a second server (Intel(R) Xeon(R) CPU E5-2650 with Nvidia Tesla K80) and a variety of workstations. Even on slightly older hardware, the results of the approach was reproduced (e.g. Nvidia GTX 960).

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

# A APPENDIX

## A.1 ARCHITECTURE OF DISCRIMINATOR AND GENERATOR

Structure of our discriminator and generator of our cGAN approach is shown in fig. 11 and 12.

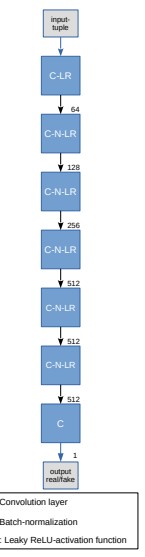

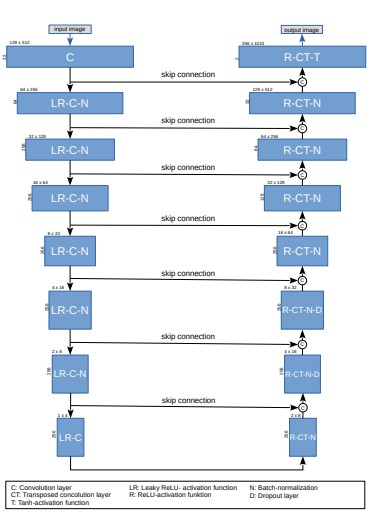

Figure 11: Structure of our discriminator.

Figure 12: Structure of our U-Net with 8 layers.

## A.2 DATASET

For an insight into the data set, please refer to fig. 13. Here, the respective images in x- and y-direction as well as the pressure field are presented at the time points $t = 0, 25, 100, 250$.

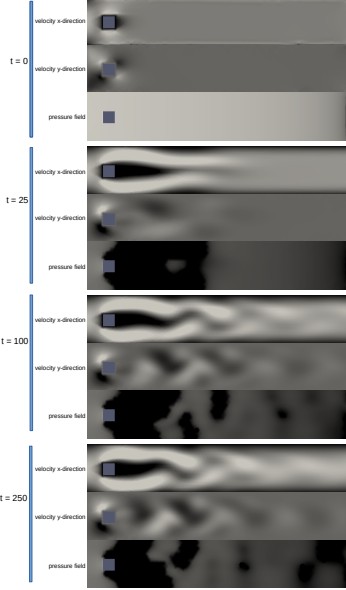

Figure 13: Example flow at $t = 0, 25, 100, 250$ with $s_{in} = 1.5625 \frac{m}{s}$.

### A.3 Test Environment

Training and testing are performed on two different environments. For training, we used the opportunity to calculate on a high-performance computer. One node consists of two Intel Xeon Gold 6230 CPUs and four NVIDIA Tesla V100. For testing and evaluating, we used our local server with two AMD EPYC 7F32 CPU and one Nvidia RTX A6000.

### A.4 Further look into the evaluation

#### A.4.1 Generator with eight layers without LSTM

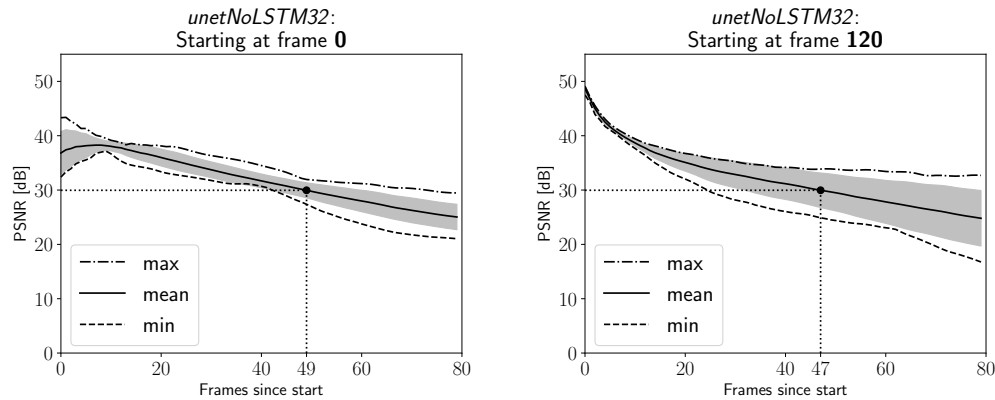

Figure 3: Performance chart of *unetNoLSTM32* with respect to PSNR. The dashed lines are highlighting the value range while the gray area shows the standard deviation around the mean value (solid line). The iteration step where the mean value falls below the value of 30 $dB$ is marked with the dotted line.

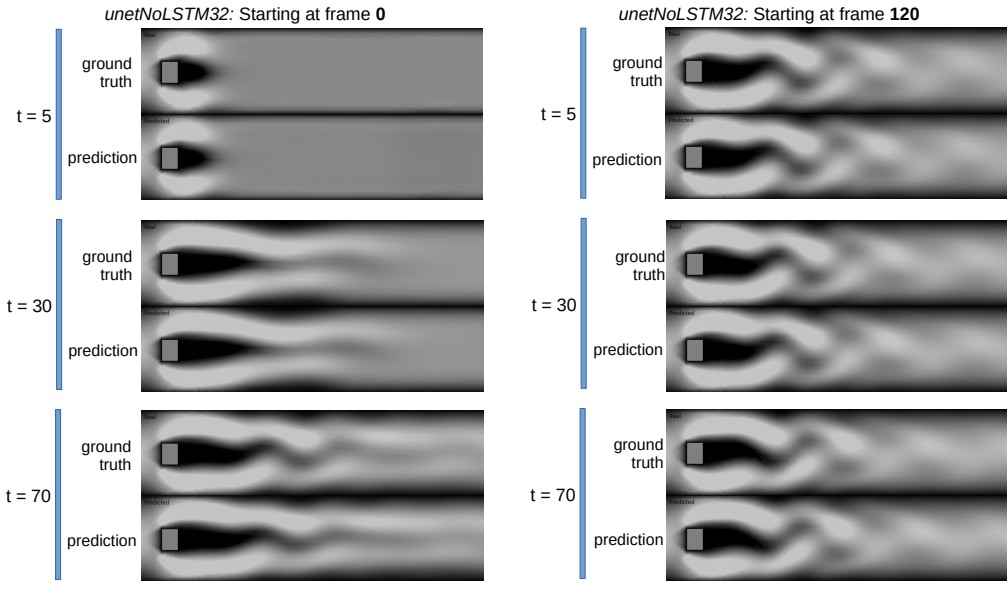

Figure 4: Predictions of *unetNoLSTM32* with starting frame zero and 120 at $t = 5, 30$ and 70 in comparison to the ground truth.

### A.4.2 GENERATOR WITH EIGHT LAYERS WITH LSTM

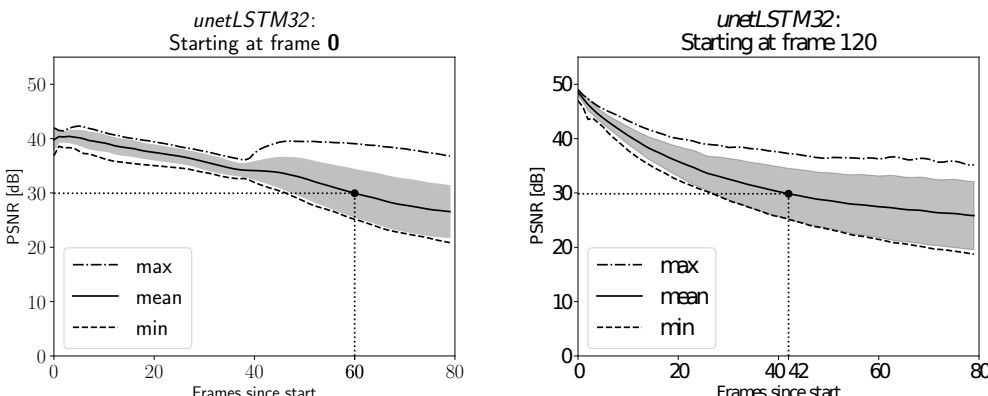

Figure 5: Performance chart of *unetLSTM32* with respect to PSNR. The dashed lines are highlighting the value range while the gray area shows the standard deviation around the mean value (solid line). The iteration step where the mean value falls below the value of $30 \ dB$ is marked with the dotted line.

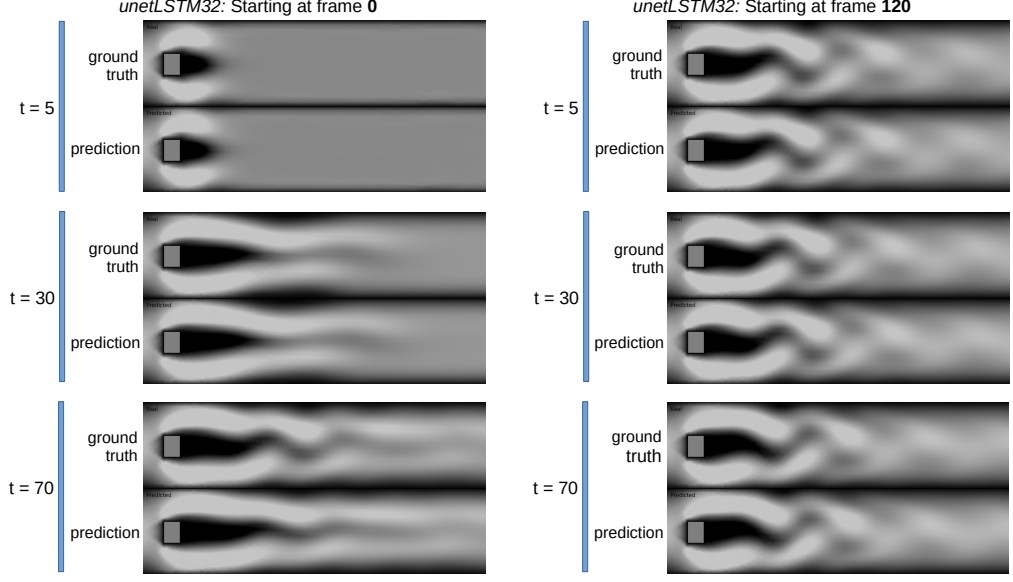

Figure 6: Predictions of *unetLSTM32* with starting frame zero and 120 at $t = 5, 30$ and 70 in comparison to the ground truth.

### A.4.3 GENERATOR WITH EIGHT LAYERS WITH LSTM WITH ADDITIONAL INPUT TO LSTM

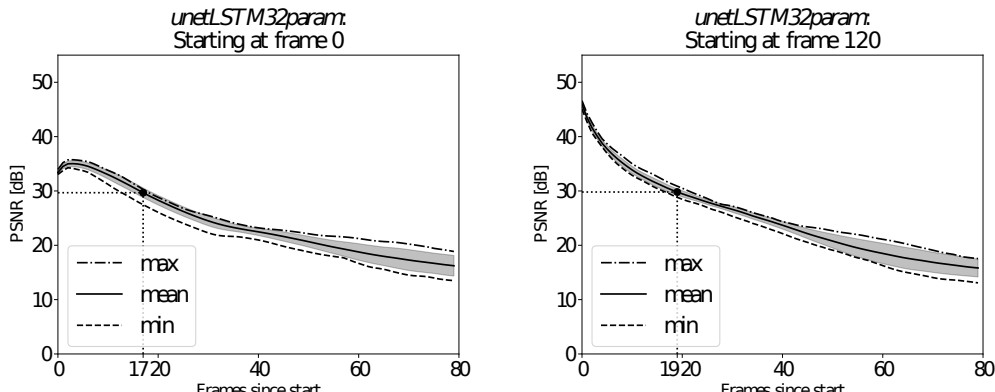

Figure 7: Performance chart of *unetLSTM32param* with respect to PSNR. The dashed lines are highlighting the value range while the gray area shows the standard deviation around the mean value (solid line). The iteration step where the mean value falls below the value of $30\ dB$ is marked with the dotted line.

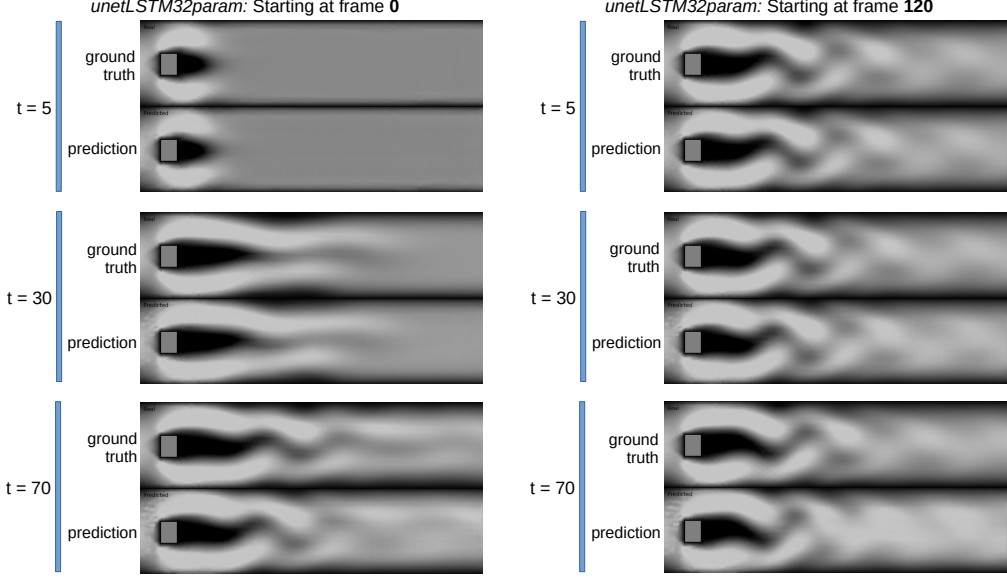

Figure 8: Predictions of *unetLSTM32param* with starting frame zero and 120 at $t = 5, 30$ and 70 in comparison to the ground truth.

### A.4.4 GENERATOR WITH EIGHT LAYERS WITH LSTM WITH MORE FEATURE MAPS

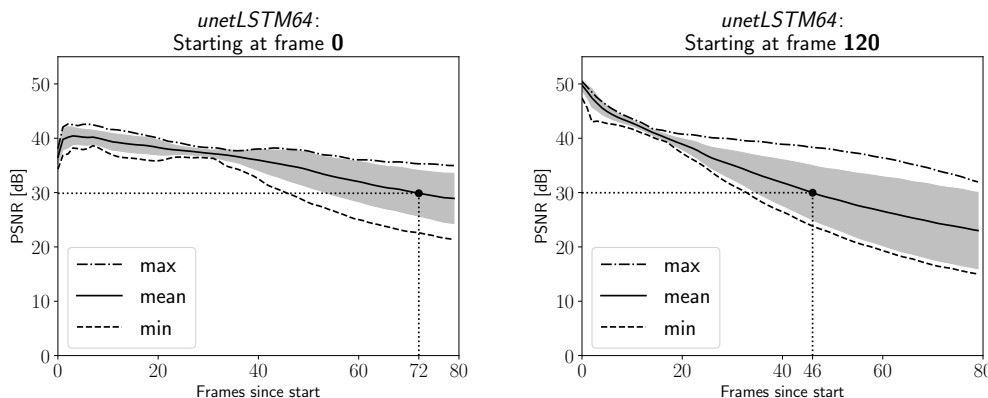

Figure 9: Performance chart of *unetLSTM64* with respect to PSNR. The dashed lines are highlighting the value range while the gray area shows the standard deviation around the mean value (solid line). The iteration step where the mean value falls below the value of $30\ dB$ is marked with the dotted line.

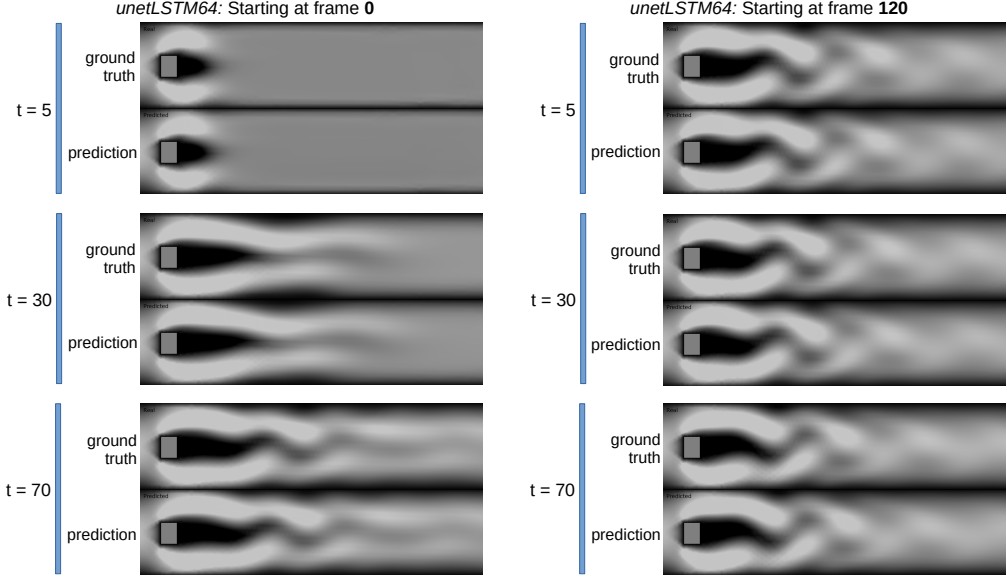

Figure 10: Predictions of *unetLSTM64* with starting frame zero and 120 at $t = 5, 30$ and 70 in comparison to the ground truth.

