# OpenReview forum: "A New Perspective on Fluid Simulation: An Image-to-Image Translation Task via Neural Networks"
_ICLR.cc/2022/Conference — ICLR 2022 Submitted_

### Official Review · Reviewer_jvzR · 2021-10-31

**Correctness:** 2
**Technical Novelty And Significance:** 1
**Empirical Novelty And Significance:** 1
**Recommendation:** 1
**Confidence:** 5

**Main Review:**

The paper uses the cGAN method in [Isola et al 2017] on image sequences generated by a fluid solver. These images correspond to 8-bit discretizations of the velocity and pressure fields.
There is no novelty in the method (this is 1:1 reimplementation of Isola et al), nor the task (Karman vortex street is a standard benchmark task for both classical, and ML-based physics prediction papers).
It's unclear to me what benefits an 'image-based' approach would bring. There are lots of papers modeling flow fields on regular grids with CNNs (e.g. de Bezenac "Deep Learning for Physical Processes" or Thuerey "Deep learning methods for Reynolds-averaged Navier–Stokes simulations"). The only difference in the encoding is that these papers don't discretize the flow fields to 8-bit values.
The paper does not contain any baseline comparisons to other methods, or further analysis.

**Summary Of The Paper:**

The paper uses a cGAN to model the fluid flow of a Karman vortex street in image-space. This is intended as a more efficient method for creating 'visual' simulations compared to classical numerical simulation.

**Summary Of The Review:**

This paper uses a standard implementation of a cGAN on a standard benchmark task for physics prediction. There is no significant innovation, comparison to any of the many other ML approaches solving this problem, or interesting analysis and discussion.
I don't think this paper brings any new insights to the community.

---

### Official Review · Reviewer_BH97 · 2021-11-01

**Correctness:** 3
**Technical Novelty And Significance:** 2
**Empirical Novelty And Significance:** 2
**Recommendation:** 3
**Confidence:** 5

**Main Review:**

The idea of fluid simulation as an image-to-image translation task is really interesting. The authors put in considerable work evaluating various architectures for this specific task, and come up with very nice looking fluid flow predictions that score well on PSNR.

Interesting the idea may be, I think the execution of the idea needs some work. The paper is missing important related work, relevant comparisons and baselines, and information about the architecture (although this info can be inferred from the supplementary material). Moreover, the experiments are conducted on a small dataset, are likely to be overfitting, and are unfair to the selected baseline. The writing in the paper is also poor.

My first concern regards baselines. FEM and LBM are not the only numerical solvers for fluid flow. There has been high demand for fast and only mildly accurate fluid simulations for a long time, mainly from video games and graphics. This body of work is missing from the related works section, and in my opinion from the comparisons and baselines of the experiments in this paper. "Real-Time Fluid Dynamics for Games" for example is a 2003 paper describing a simple fluid dynamics solver that is capable of producing Kelvin-Helmholtz and Rayleigh-Taylor instabilities, in addition to Karman vortex sheets, in real-time on a 2000's CPU. Given that these "fast and only mildly accurate" methods exist, I feel the authors must justify why using a neural network to simulate fluids would be useful or interesting. Are they faster and more accurate than existing methods? Are ouputs more realistic looking?

The experiments in the paper are also poorly described and executed. The architecture of the U-net is described in detail, yet I could not find a detailed description of how an LSTM was combined with the U-net and the GAN setup, and how exactly the inflow velocity was given to the U-net. I also could not find info on the training and test data, except that there were 33 simulations, split 80-20, exclusively of the vortex shedding phenomena. With such a small dataset size, there is likely overfitting occuring (a possibility described but not addressed by the authors: "overfitting effect is probably certainly present" in section 3.4). I am further concerned that overfitting is occurring because I do not believe the network is ever given enough information to fully predict fluid flow perfectly. The Navier-Stokes equations require both pressure and velocity to be known globally, yet the network is only given pressure and local inflow velocity. The trained network may simply be a "Karman vortex sheet simulator."

Finally, the results of the experiments are lacking nuance. The authors claim their method trades off accuracy for speed, yet there is no explicit analysis of this tradeoff. The authors compare against a single FEM solver, which is slower but more accurate than the proposed neural network approach, but it is not established quantitatively how much more accurate. The baseline itself is rather unfair as well, as the method they compare against is actually a multiphysics tool which can solve general BVPs from electrodynamics, fluid dynamics, heat transfer, and acoustics, whereas the proposed method only works to simulate Karman vortex sheets for very specific boundary conditions.

Additional Questions:

- What is the LSTM architecture?
- How is the inflow velocity given to the network?
- How large is the dataset?
- Can we be sure there's no overfitting?
- What is the "approximation property"? (sec 1.1)
- What is the "false streaming data"? (sec. 2.1)

Additional Feedback

- Including the image processing time of paraview seems unfair
- PSNR likely needs no elaboration in sec. 2.1, as it is a widely used metric
- In section 2.2, last sentence of second paragraph seems to imply cGANs are very different from GANs.
- There is, in my opinion, a disproportionately large emphasis on the architecture used (number of features, layers, LSTMs). I think just presenting the best results could be sufficient

- The writing is sort of poor at points in the paper. As a result, it's sometimes hard to understand the ideas being communicated. The paper will probably benefit greatly from a round of revision and editing.
- Various typos/poor writing:
    - "manly" -> "mainly" (sec. 1.2)
    - Many different spellings of u-net ("u-net," "UNET," "U-NET")
    - eqn 3 typo "A_G"
    - "Karman vortex street" (sec 3.1)

**Summary Of The Paper:**

The authors pose fluid simulation as an image-to-image translation task. From this perspective, approximating fluid flow using a cGAN can potentially improve speed (at the expense of accuracy) over FEM methods. This can be useful in situations where accuracy can be traded off for speed (e.g. video games). The authors show a U-net can produce good future predictions for Karman vortex sheets, and is faster than an off the shelf multiphysics simulator. In addition the authors ablate various architectural design choices.

**Summary Of The Review:**

The idea is interesting, but the execution of the paper is poor. The experiments lack rigor, there is likely massive overfitting, and the baselines are incomplete and weak. The writing in the paper could also use some work.

---

### Official Review · Reviewer_9rKJ · 2021-11-01

**Correctness:** 3
**Technical Novelty And Significance:** 1
**Empirical Novelty And Significance:** 1
**Recommendation:** 1
**Confidence:** 5

**Main Review:**

The proposal to treat simulation and visualization jointly by considering the whole problem in image space is a novel aspect of the paper, which the authors clearly note. However, this is never motivated well and reading the paper, I was not convinced that it makes sense. In fact, the authors themselves note clear drawbacks of this approach ("the very limited resolution of the color spaces alone results in high errors"), and other motivating factors are unclear ("the Elmer FEM solver produces a non-intuitive data format that must be processed by Paraview"; what is non-intuitive about it? what about simulator frameworks that have an integrated visualization component?)

The paper does not cite or compare to relevant prior work. There is now a large body of literature using neural networks as a replacement for or extension of fluid simulators, including works utilizing GANs and various convolutional architectures and black-box approaches attempting to model the flow field without relying on knowledge of the underlying physics (e.g. doi:10.1109/ICTAI50040.2020.00057, doi:10.1017/jfm.2019.700), This appears to have been completely ignored here. Papers investigating simulation accuracy from the point of view of a human observer (e.g. https://arxiv.org/abs/1907.04179) are also not discussed.

The authors state that a motivating factor for their project is speed, but the proposed approach does not really deliver on this front. In Table 1, CPU cost of inference (not taking training into account) is always comparable to a FEM solver, and GPU speedup is not particularly relevant due to the lack of a GPU-accelerated FEM baseline. The authors also state that "there is no mathematical way to decrease this correctness in cases where only approximate predictions are needed.". It is unclear what counts as "mathematical" in this context, but this seems too broad a statement, and approaches like subgrid models or adaptive mesh refinement could be discussed in this context.

The paper contains very little discussion about generalization -- there is just a brief mention of experiments with flow around a cylinder instead of a rectangle, but no further details. Insufficient information about the training and testing set makes it impossible to say how well the model works e.g. as a function of the Reynolds number. In section 3.4, variations in the placement of the rectangular obstacle are discussed, but again no details are provided.

Technical comments:
- "streaming data" is mentioned multiple times in the text, but it's unclear what this means. Is it the velocity field?
- The text mentions "33 random-picked simulations" for training. How were the parameters varied? Which Reynolds numbers were covered?
- The details of the FEM simulation are missing. What mesh resolution & type and time step size were used?


**Summary Of The Paper:**

The paper casts the problem of 2d fluid flow simulation as an image to image-to-translation task. A cGAN with standard architectures (U-net generator, PatchGAN discriminator) is trained to advance the visualization of the simulation to the next time step, and an extension with an LSTM block is explored. The model is evaluated in an autoregressive setting on the problem of fluid flow around a rectangle, and the results are evaluated with image metrics (PSNR).

**Summary Of The Review:**

The lack of comparison to prior work, the usage of non-standard metrics, and lack of novelty (beyond the general framing, which I found poorly motivated) make it impossible for me to recommend acceptance of this paper in the present form.

---

### Official Review · Reviewer_1Jbg · 2021-11-06

**Correctness:** 1
**Technical Novelty And Significance:** 1
**Empirical Novelty And Significance:** Not applicable
**Recommendation:** 3
**Confidence:** 4

**Main Review:**

Although the authors reformed the problem as image translation, using a cGAN (instead of GAN, or other architecture) to generate PDE's result seems to be a trivial idea. Indeed, explicitly declaring the problem as image translation could be a new perspective. Still, it did not provide many novel insights, nor did it help derive a novel machine learning method that outperforms the practices in other related works. In addition, there are other critical issues in the paper.

The paper is poorly written with fancy words. In the abstract, the authors claimed, "we developed an approach based on Neural Networks that is much less time-consuming but nearly as accurate as the numerical model for a human observer." What kind of human observer? Someone with sharp eyes? On a monitor or a piece of paper? Such statements are vague and erratic in the context of solving a PDE.

The PSNR is just a new form of loss, and using a PSNR as the loss is irrelevant to other fancy claims (e.g., human observers). The paper also didn't provide a convincing reason that PSNR is more reliable than different loss formulations.

Besides, the method is poorly validated. The authors only test their approach on a single example with a fixed resolution, showing no comparison with other state-of-the-art methods. Hence I am not convinced that the method in this paper is superior to any other related works in solving PDEs. In the extent of "a human observer", the authors didn't show any user studies to justify their result is indiscernible to sharp eyes.


**Summary Of The Paper:**

This paper reform the task to solve a PDE into an image translation problem and used cGAN to solve the PDE.

**Summary Of The Review:**

I am not convinced that this paper is contributive (against the bar of ICLR) to either the machine learning or numerical analysis communities. Hence I would oppose its acceptance.

---

### Decision · Program_Chairs · 2022-01-20

**Decision:**

Reject

**Comment:**

The paper formulates fluid simulation as an image to image prediction task and proposes to solve the problem using a cGAN formulation. The objective is to develop fast approximate solutions for the modeling of fluid dynamics, here Navier Stokes for incompressible flows. The images correspond to the discretization of velocity and pressure fields. Experiments are performed on a simulation for a Karman vortex street.

All the reviewers expressed concerns w.r.t. the absence of references and comparisons with closely related work in the recent but abundant literature on NN for modeling PDE dynamics, the lack of novelty and the insufficient experimental design, description and discussion.